# Student Behavior Detection in the Classroom Based on Improved YOLOv8

**DOI:** 10.3390/s23208385

**Published:** 2023-10-11

**Authors:** Haiwei Chen, Guohui Zhou, Huixin Jiang

**Affiliations:** 1School of Computer Science and Information Engineering, Harbin Normal University, Harbin 150025, China; chenhaiweibooty@gmail.com; 2School of Life Sciences and Technology, Harbin Normal University, Harbin 150025, China; jianghuixinchen@gmail.com

**Keywords:** classroom behavior detection, YOLOv8, EMA, MHSA

## Abstract

Accurately detecting student classroom behaviors in classroom videos is beneficial for analyzing students’ classroom performance and consequently enhancing teaching effectiveness. To address challenges such as object density, occlusion, and multi-scale scenarios in classroom video images, this paper introduces an improved YOLOv8 classroom detection model. Firstly, by combining modules from the Res2Net and YOLOv8 network models, a novel C2f_Res2block module is proposed. This module, along with MHSA and EMA, is integrated into the YOLOv8 model. Experimental results on a classroom detection dataset demonstrate that the improved model in this paper exhibits better detection performance compared to the original YOLOv8, with an average precision (mAP@0.5) increase of 4.2%.

## 1. Introduction

The big data smart classroom has become an inevitable trend in the future development of education [1]. At this stage, the observation and recording of students’ classroom learning behavior mainly relies on the human supervision of the classroom site by the lecturer, as well as the assessment of learning behavior through video data at a later stage. However, there are two problems with these two methods: first, the lecturer is distracted, which reduces the efficiency of the lecture; second, it is not possible to comprehensively and accurately record the learning behavior of all students. Therefore, there is a growing expectation of the use of deep-learning and computer-vision techniques to analyze students’ classroom behaviors. Using computer-assisted teaching to automatically detect and analyze students’ classroom behaviors has also become a hot research topic in smart education [2,3].

There are a total of three classes of algorithms currently used for student behavior detection: video-action-recognition-based [4], gesture-estimation-based [5], and object-detection-based [6]. Identifiable persistent behavior for student classroom behavior detection is based on video action recognition; however, this requires the labeling of a large number of samples. For example, the AVA dataset [7] for SlowFast [8] detection labels 1.58 million samples. Moreover, video behavior recognition detection is still immature; for example, Charades [9] and Kinetics400 [10] show that some actions can sometimes be judged only by context or scene. Algorithms based on pose estimation describe human behaviors by obtaining information about the position and movement of each joint of the human body but are not applicable to behavior detection in crowded classrooms. Considering the current challenges, object-detection-based algorithms are a promising solution. In fact, great breakthroughs in object-detection-based algorithms have been made in recent years, e.g., YOLOv8 [11]. Therefore, this paper uses an algorithm based on object detection to analyze student behavior in the classroom.

The main challenge in object detection is to recognize multiple objects of different classes in an image and to provide each object with positional information about its bounding box [12]. In the past, object detection was usually based on hand-designed features and traditional machine-learning methods such as SVM (Support Vector Machine) [13] and Haar cascade [14]. However, with the development of deep learning, especially the rise of convolutional neural networks (CNNs), object detection has made tremendous progress. It has received more attention due to its impressive detection results on public datasets.

The real-classroom dataset is very different from the public dataset, and classical methods do not perform well in real classrooms. One representative problem is the high scale variation between different locations, e.g., the proportion difference between students in the front row of the classroom (approximately 40 × 40 pixels) and students in the back row (approximately 200 × 200 pixels) reaches up to 25 times. What is even worse is that, compared to the most popular object-detection dataset, MS COCO [15], the occlusion between students in the real-classroom dataset is very severe. In addition, external distractors, such as environment, angle, and other people, in real classrooms have an impact on student behavior.

As shown in Figure 1, a multi-scale visual pattern occurs with the size of the characters in the student classroom images. First, students may appear in the same image at different sizes, e.g., a student in the front row and a student in the back row. Second, information about the underlying context of an object may be more important than information about the object itself in the recognition process, e.g., a person standing at a podium lecturing is more likely to be a teacher than a student. Third, sensing information at different scales is critical for tasks such as target detection [16]. Therefore, it is crucial to design the use of multi-scale stimuli in student behavior detection.

Feature fusion is one of the common methods used to solve multi-scale target detection [17]. Most of the structures of convolutional networks use top-down data flow: the deeper the convolutional layer the larger its receptive field, and the deeper the features the richer the semantic information carried. The smaller the scale of the feature spectrum, the smaller the pixel area occupied by small objects, and, as the receptive field increases, the features of small objects will be blended with more background information, which is not conducive to the detection of small objects. Feature fusion adds high-level semantic information to the low-level feature spectrum through top-down feedback branching, so that the high-resolution features at the low level can also obtain rich semantic information, enabling the low-level features to contain both detail information and category information, in order to improve the detection performance of small targets. The feature pyramid network FPN [18] is a standard component of existing target detection networks, which are used to generate multiple feature spectra at different scales, e.g., MLFPN [19], AugFPN [20], and YOLOv8, which rely on multi-layer feature spectra to detect targets at different scales in different combinations. MLFPN generates the base features by fusing the multi-level features, and then combines the base features in a hierarchical way. It then inputs the base features into several codec layers as a cascade structure, each of which outputs a series of multi-scale features, and then, finally, collects the features of the same scale at each level of output by all the decoders to construct a multi-level feature pyramid for target detection. AugFPN fuses features after RoI pooling using spatial attention spectra. It is also important to address multi-scale target detection at a fine-grained level [21]. Res2Net achieves this by constructing hierarchical residual-like connections in a single residual block to represent multi-scale features at a fine-grained level, increasing the range of sensory fields at each network layer. The efficient multi-scale attention (EMA) module reshapes some of the channels into batch dimensions and groups the channel dimensions into multiple sub-features, resulting in an even distribution of spatial semantic features within each feature group. Specifically, in addition to encoding global information to recalibrate the channel weights in each parallel branch, the output features of the two parallel branches are further aggregated through cross-dimensional interactions to capture pixel-level pairwise relationships.

In this paper, we simultaneously draw on the above two methods for solving multi-scale target detection, using YOLOv8 as the basic network framework, upon which we embody the ideas of Res2Net and introduce the efficient multi-scale attention module (EMA) for cross-space learning into the neck network. YOLOv8 adopts multi-level feature fusion and extracts feature maps at different levels through pooling and step-size adjustment. These feature maps are then fused to obtain richer perceptual information and improve the accuracy of object detection.

The basic idea of the attention mechanism is to enable an algorithm or model to learn a strategy to adaptively allocate information-processing resources, so that the network can filter unimportant information like the human brain, and thus allocate more information-processing resources to features with discriminative information. Inspired by Transformer [22], this mechanism flexibly captures spatially different local saliency of the whole image and generates multiple attention maps for a single image from different aspects. MHSA is a deep learning model based on the self-attention mechanism, which was first proposed by Vaswani et al. in 2017 [23]. With MHSA, noisy or unimportant regions can be cut out and key local feature information can be highlighted. MHSA can help the classroom detection model to better capture the key information of the target character in the unobscured region and avoid the information in the occluded region. Therefore, the multiple-head self-attention mechanism (MHSA) is introduced in the student classroom detection model in this paper.

Classroom student behavior detection is crucial for classroom teaching and learning as well as healthy student development [24]. Therefore, this paper aims to enhance YOLOv8 by improving its ability to extract multi-scale spatial features from feature maps. The goal is to enhance detection accuracy, improve teacher efficiency during lectures, and reduce the frequency of teacher distractions. To enhance the performance of YOLOv8, we have made several improvements to the model. The main contributions of this paper are as follows:Based on the idea of multi-scale structure in Res2Net, the C2f_Res2block module is proposed by integrating the Res2Net module therein with the C2f module in YOLOv8 [25]. This module improves the performance and robustness of the whole-target detection model.We introduce the newly released multi-scale attention module EMA [26] to merge with the YOLOV8 backbone to further improve the model’s stimulation of targets at different scales.Finally, inspired by Transformer, we add a module with the multiple-head self-attention mechanism (MHSA) to the YOLOv8 neck module. MHSA can help the classroom detection model to better capture the key information of the target character in the unobscured region and avoid the information in the occluded region.We tested the improved YOLOv8 detection network framework on SCB-Dataset, and its mAp0.5 was improved by 4.2% over the original YOLOv8.

This document’s remaining sections are structured as follows: Section 2 provides an overview of the relevant technologies designed in the paper, while Section 3 introduces our methodology. In Section 4, we describe the related experiments and discuss the research findings. Finally, Section 5 summarizes the conclusions and offers suggestions for future work.

## 2. Related Work

### 2.1. YOLOv8 Framework Review

YOLOv8 is a target detection model (the basic architecture of YOLOv8 is shown in Figure 2). It is the latest version of the YOLO series of models. YOLOv8 adopts an anchor-free-based detection approach, which means that it directly predicts the target’s center point and width-to-height ratio instead of predicting the position and size of the anchor box. This approach can reduce the number of anchor boxes and improve detection speed and accuracy. The principle can be divided into two parts: feature extraction and target detection. However, in the actual detection of real classrooms, there are still some deficiencies in dealing with the problems of dense video image objects, mutual occlusion between members, and multi-scale detection of objects. To address these issues, this paper utilizes YOLOv8 to make a series of improvements.

### 2.2. Res2Net

Multi-scale features have always been important in detection tasks. Since the proposal of null convolution, the multi-scale pyramid model, built on the basis of null convolution, has achieved milestone results in detection tasks. The object’s information obtained under different receptive fields varies. A small receptive field may capture more details of the object, which is also very beneficial for detecting small targets. In contrast, a large receptive field can capture the overall structure of the object, making it convenient for the network to locate the object’s position. The combination of details and position can better extract information about objects with clear boundaries. Therefore, models that incorporate multi-scale pyramids often achieve very good results.

As shown in Figure 3, feature $k_{2}$ is fed into the processing stream where $x_{3}$ is located after a 3 × 3 convolution. $k_{2}$ is again optimized for information by the convolution of 3 × 3, and two 3 × 3 convolutions are equivalent to a 5 × 5 convolution. Then, $k_{3}$ is taken for granted with the fusion of the processed features of the 3 × 3 receptive field and the 5 × 5 receptive field. By analogy, a 7 × 7 receptive field is applied to $k_{4}$. In this way, Res2Net, when used for detection tasks, can extract multi-scale features to improve the accuracy of the model. In this paper, we utilize this advantage by combining it with the original C2f module in YOLOv8 to propose a new C2f_Res2block module, which improves the model’s ability to extract multi-scale space in the feature map.

### 2.3. Efficient Multi-Scale Attention

The attention at multiple scales (EMA) [26] module reshapes some of the channels into batch dimensions and groups the channel dimensions into multiple sub-features so that the spatial semantic features are evenly distributed within each feature group. Specifically, in addition to encoding global information to recalibrate the channel weights in each parallel branch, the output characteristics of the two parallel branches are further aggregated through cross-dimensional interactions to capture pixel-level pairwise relationships.

The core idea of the EMA attention mechanism is to introduce the concepts of excitation and modulation to the traditional attention mechanism. The excitation mechanism calculates the importance of each part of the input data for the task at hand, while the modulation mechanism adjusts the weights of different parts to achieve better model performance. The excitation mechanism determines the importance of each part by calculating the similarity between the input data features and parameters. Specifically, it generates a similarity matrix by computing the inner product of the input data with the parameters. Each element in this matrix represents the similarity between a part of the input data and the parameter. A higher similarity indicates that the part is more important for the current task. Next, the modulation mechanism adjusts the weights of each part based on the similarity matrix calculated by the excitation mechanism. The modulation mechanism can be implemented in various ways, with common approaches including normalization using the softmax function. By applying softmax normalization to each row of the similarity matrix, a weight vector is obtained, which represents the importance of each part for the current task. Then, a weighted summation operation is applied to the input data using the weight vector to obtain a weighted representation of the network for the input data.

The advantage of the EMA attention mechanism is its ability to extract important information relevant to the current task from the input data, thereby reducing interference from irrelevant information for the model. Therefore, this attention mechanism is introduced in this paper to enhance the detection accuracy of the model.

### 2.4. Transformer Detection Algorithm

The application of Transformer in target detection is based on its powerful self-attention mechanism and its ability to capture sequential information. Transformer’s self-attention mechanism is well-suited for capturing global dependencies in images. In particular, the multi-headed attention mechanism (MHSA) in the model, which excels at handling complex relationships, addressing long-distance dependencies, and capturing multi-level relationships, enables the model to better understand the connections between upper and lower graphics. Therefore, this paper introduces the multi-headed attention mechanism (MHSA) into the YOLOv8 classroom detection model.

## 3. Methodologies

### 3.1. Overall Framework

This section provides an overview of the entire framework, as illustrated in Figure 4. In this paper, we propose an improved YOLOv8 model for classroom behavior detection. The original training images are processed through this enhanced YOLOv8 model to detect and recognize students’ classroom behavior. Specific methods are presented in Section 3.2.1, where we introduce the Res2Net module and the C2f_Res2block module, which replaces all the original C2f modules. In Section 3.2.3, we incorporate the EMA attention mechanism into the backbone. Furthermore, in Section 3.2.3 we introduce the multi-headed attention mechanism (MHSA) into the backbone.

### 3.2. Improved YOLOv8

To enhance detection accuracy, we propose an improved YOLOv8-based model for classroom student behavior detection. In this endeavor, we have designed a new object detection framework by combining the Res2Net module with the C2f module to create the C2f_Res2block module. We replace all the C2f modules in the original YOLOv8 with this module and introduce the application of the efficient multi-scale attention module with cross-spatial learning into the neck network. The overall structure of the enhanced detection framework is depicted in Figure 5, which illustrates the differences between the classic YOLOv8 and the improved YOLOv8. In the figure, the yellow background box region represents the added EMA block, the orange background box region represents the added multi-headed attention (MHSA) module, and the green highlighted square represents the replaced C2f_Res2block module. Experimental results verify that the improved YOLOv8 framework exhibits superior performance and detection accuracy.

#### 3.2.1. C2f_Res2block Module Proposed in This Study

As shown in Figure 6, this module is compared to the C2f module, wherein the backbone module is replaced by the Res2Net module. With this enhancement, the model is capable of extracting a wider range of multi-scale features.

As shown in Figure 7, when ‘Shortcut’ is set to False, it is directly output through a 1 × 1 convolution layer. When ‘Shortcut’ is set to True, it is fused with the original input features after a 1 × 1 convolution and then input. To better integrate information from different scales, we combine all the segmentation information and pass it through a 1 × 1 convolution. This segmentation and concatenation strategy enhances convolution and processes the features more efficiently. To reduce the number of parameters, we skip the convolution in the first segmentation, which can also be viewed as a form of feature reuse. The Res2Net module outperforms the backbone module in multi-scale detection.

As shown in Figure 7, the Res2Net module, after 1 × 1 convolution, divides the input feature x into k subsets of feature maps, each of which i has the same spatial size compared to the input feature maps, but with the number of channels 1/k. Except for i=0, each i has a corresponding 3 × 3 convolution, denoted by $G_{i}()$. We denote the $G_{i}()$ output by $y_{i}$. The feature subset i is summed with the output of $G_{i-1}()$, which is then input to $G_{i}()$. Thus, $y_{i}$ can be written as:(1)$$y_{i}=\left\{\begin{array}{l} i\\ G_{i}(i)\\ G_{i}(i+y_{i-1}) \end{array}\right.\begin{array}{ccc} & i=1; & \\ & i=2; & \\ & 2<i\leq k; & \end{array}$$

#### 3.2.2. Neck Network with EMA

The improved YOLOv8 neck network model is shown in Figure 5. In this paper, the EMA module is added to the beginning of the YOLOv8 neck framework, which does not change the size of the feature vectors. The EMA is the core module of the improved YOLOv8 as shown in Figure 8. For any given input feature map X∈ℝC×H×W, EMA will divide X into G sub-features across the channel dimension directions for learning different semantics, where the grouping can be represented by X=[X0,X1,…,XG−1],Xi∈ℝC//G×H×W. In order to better collect multi-scale spatial information, we change the original 3 × 3 branching to 5 × 5 branching, which increases the sensory field of the model. EMA utilizes three parallel paths to extract the attention weight descriptors of the grouped feature maps. Two of the parallel paths are 1 × 1 branches, and the third path is a 5 × 5 branch.

EMA provides a method for aggregating information across spaces in different spatial dimensional directions for richer feature aggregation. It is worth noting that we still introduce two tensors here, where one is the output of the 1 × 1 branch, and the other is the output of the 5 × 5 branch. We then encode the global spatial information in the output of the 1 × 1 branch using 2D global average pooling, and the output of the smallest branch is directly converted to the shape of the corresponding dimension, i.e., ℝ11×C//G×ℝ3C//G×HW [27], prior to the joint activation mechanism of the channel features. The formula for the 2D global pooling operation is
(2)zc=1H×W∑jH∑iWxc(i,j)
where zc is the output associated with the c-th channel. The aim is to encode global information and model long-range dependencies.

#### 3.2.3. Neck Network with MHSA

The improved YOLOv8 model is shown in Figure 5. In this paper we add it to the front of the detection head, which learns and captures contextual information more efficiently and does not change the size of the feature vector.

As shown in Figure 9, the input size of the multi-head attention module is H×W×d, where H, W, and d represent the height and width of the feature matrix along with the size of an individual label. We perform a 1 × 1 convolution operation on the input data to obtain query encoding, key encoding, and value encoding, and the query encoding and key encoding matrices are multiplied to obtain the content information. Then, matrix multiplication with the value encoding is carried out after softmax operation to obtain the output results. Since there is no positional embedding in the non-native stratum, it is removed when the original multi-head attention mechanism is introduced. This means that the MHSA used in this experiment only considers the content information and not the content location.

## 4. Experiments

### 4.1. Experimental Details

#### 4.1.1. Datasets

In this study, we used a publicly available student classroom behavior dataset (SCB-Dataset) to evaluate the effectiveness of the classroom behavior detection method we proposed [28]. The SCB-Dataset includes 18.4 thousand labels and 4.2 thousand images covering three behaviors: raising hands, reading, and writing. An example of the dataset is shown in Figure 10. We evaluated the performance of the whole framework by dividing it into training and validation sets at a ratio of 4:1.

#### 4.1.2. Assessment of Indicators

In order to comprehensively and objectively assess the performance of the proposed model, we utilized the mean average precision (mAP) to measure the accuracy of the model and evaluate the object detection results. TP represents the true positives (the number of target frames that are correctly predicted to be in the positive category), FP represents the false positives (the number of target frames that are incorrectly predicted to be in the positive category), and FN represents the false negatives (the number of target frames that are actually in the positive category but are incorrectly predicted to be in the negative category).

Precision is the ratio of the number of target boxes correctly predicted by the model as positive categories to the number of all target boxes predicted by the model as positive categories and is defined as:(3)$$\mathrm{Precision}=\frac{\mathrm{TP}}{\mathrm{TP}+\mathrm{FP}}$$

Recall is the ratio of the number of target frames correctly predicted as positive categories by the model to the number of target frames in all actual positive categories and is defined as:(4)$$\mathrm{Recall}=\frac{\mathrm{TP}}{\mathrm{TP}+\mathrm{FN}}$$

AP is the area under the precision–recall curve and represents the average precision of the model at different recall rates. It is defined as:(5)$$\mathrm{AP}=\int_{0}^{1}\mathrm{PRdr}$$

mAP (mean average precision) is a comprehensive metric used to assess the performance of object detection models across multiple categories. It calculates the average precision (AP) for each category and then takes the average of these AP values to gauge the model’s performance.
(6)$$\mathrm{mAP}=\frac{1}{C}\sum_{i=1}^{C}AP_{i}$$
where C represents the number of categories in the dataset. The higher the mAP value, the better the model’s performance.

#### 4.1.3. Experimental Setup

The experimental environment of this study included training the model on NVIDIA GeForce RTX 3080 using GPU drivers with Ubuntu 20.04. The environment used was Python 3.8.16 and torch 1.13.1 + cu116. All experiments described in this article were set up to train for 400 epochs, and training was stopped early when there was no significant improvement in average accuracy after 50 epochs. Training was performed using the YOLOv8x model from the YOLOv8 family, with a batch size of 4, subject to GPU memory constraints. The learning rate used during model training was 0.01, with an SGD momentum of 0.937 and an optimizer weight decay of 0.0005. All other training parameters were set to the default values of the YOLOv8 network.

### 4.2. Experimental Design

To quantitatively assess the performance of the proposed overall framework, we conducted tests on the SCB-Dataset using the introduced object detection framework. We performed ablation experiments to evaluate the importance of the C2f_Res2block, EMA, and MHSA modules within the model, gaining insights into their impact on the model’s performance. To demonstrate the versatility of our proposed model, we conducted experiments on different datasets. Additionally, we compared our proposed model to state-of-the-art object detection frameworks, providing evidence that the object detection framework presented in this paper outperforms other popular object detection frameworks in terms of accuracy.

### 4.3. Experimental Results Obtained on SCB-Dataset Using an Improved Version of YOLOv8

In this paper, the model training is set to 400 epochs, and when the average accuracy does not significantly improve, the program stops the training automatically. After 229 epochs of training, the improved model achieved training results on the SCB-Dataset. Different performance metrics for the training and validation sets are shown in Figure 11.

The first three columns depict the box loss, object loss, and classification loss of the improved YOLOv8 model. The three curves in the first three columns illustrate the loss trends, with the X-axis representing the temporal progression on the training set and the Y-axis representing the overall loss values. As can be seen from the curves, the overall loss values continue to decrease and eventually stabilize as the training progresses. These results indicate that the proposed improved YOLOv8 model exhibits good fitting performance, high stability, and accuracy. The last two columns represent PR curves, with the X-axis denoting training time and the Y-axis denoting precision and recall. These curves illustrate the assessment of object detection performance as the confidence threshold changes: the closer the curve values are to 1, the higher the model’s confidence. From Figure 11, it can be observed that the proposed improved YOLOv8 model is effective.

Figure 12 shows the confusion matrix for our proposed improved YOLOv8x model, describing its predictive accuracy across three categories of student classroom behaviors in the dataset and illustrating the relationships between predictions. In the figure, rows represent true labels, columns represent predicted categories, and the diagonal elements represent the correct detection rates. It can be observed that our model achieves high accuracy in each category.

Figure 13 illustrates the PR curves of the proposed model. It can be observed that the rate of change in precision increases as recall increases. From Figure 13, it is evident that the PR curves of the proposed model are close to the upper right corner, indicating the high recall and precision of the proposed framework. The area under the PR curves is relatively large, suggesting good model performance. Furthermore, the PR curves are also smooth, indicating a relatively stable relationship between recall and precision in our proposed model.

### 4.4. Ablation Experiments

We performed ablation experiments that assessed the validity and reliability impact of the improved schemes on the YOLOv8 aspects and that also assessed their impact on the improved performance by selectively removing these improvements. Table 1 and Figure 14 provide the results of the experiments on the SCB-Dataset.

In Table 1, the first row represents the detection results of the original YOLOv8 network, which serves as the baseline for this experiment. The second row illustrates the results after incorporating the proposed C2f_Res2block module. This module captures multi-scale features at a finer granularity and extends the receptive field of each network. We observe that the mAP@0.5 and mAP@0.5:0.95 have improved by 3.2% and 3.3%, respectively, compared to the original YOLOv8. The third row depicts the results after introducing the EMA module, which divides the channel dimension into multiple sub-features to evenly distribute spatial semantic features within each feature group. This enhances multi-scale representation capabilities. Notably, the mAP@0.5 and mAP@0.5:0.95 have improved by 2.4% and 1.7%, respectively, compared to the original YOLOv8. The fourth row presents the results with the inclusion of the MHSA module, which can extract both local and global information from input data, aiding in addressing long-range dependency issues. When compared to the original YOLOv8, the mAP@0.5 and mAP@0.5:0.95 have increased by 3.1% and 1.6%, respectively. The fifth row displays the experimental results when both the C2f_Res2block and EMA modules are used simultaneously. In comparison to the baseline, the mAP@0.5 and mAP@0.5:0.95 have improved by 3.7% and 2.6%, respectively. The sixth row showcases the final results of our student classroom behavior detection model, demonstrating an increase of 4.2% and 3.5% in mAP@0.5 and mAP@0.5:0.95, respectively. These experimental findings underscore the conclusion that our improvements to YOLOv8 have indeed elevated the detection accuracy.

### 4.5. Comparative Experiments

#### 4.5.1. Comparison of Results of Different Datasets with Experiments

To assess the model’s generalization ability, this paper conducted comparative experiments using the publicly available dataset, CrowdHuman. This dataset shares similarities with the classroom behavior detection dataset, as both feature densely populated scenes, multi-scale objects, and instances that may occlude each other [29]. CrowdHuman comprises 15,000, 4370, and 5000 images for training, validation, and testing, respectively. The training and validation subsets collectively contain 470,000 individuals, with an average pedestrian count of 22.6 individuals per image. The improved YOLOv8 model proposed in this paper was trained and tested alongside the original YOLOv8 model, and the results are presented in Table 2.

From these results, it can be observed that the improved YOLOv8 model in this study achieved favorable outcomes on the CrowdHuman dataset. It exhibited a 2.1% increase in mAP@0.5 compared to the original YOLOv8 and a 1% increase in mAP@0.95. This indicates that the improved model in this research possesses strong generalization capabilities and can be applied effectively in dense, multi-scale, and occlusion-prone classroom scenarios.

#### 4.5.2. Comparison of Results of Different Models with Experiments

In this section, we present our analysis of our proposed model in comparison with the current most popular and state-of-the-art methods. All experiments were performed on the SCB-Dataset with the original YOLOv8x as the benchmark. The experimental results are shown in Table 3 and Figure 15. The mAP@0.5 accuracy of the improved YOLOv8 student classroom testing model reaches 76.3%, which is significantly higher than that of the original YOLOv8 model. In addition, it is 5.9% higher than YOLOv5x, 5.1% higher than YOLOv8l-MHSA-C2f-Cn [30], and 3% higher than Faster-Rcnn [16]. The experimental results show that the model proposed in this paper has strong performance.

To illustrate the higher accuracy of the model proposed in this study, we selected some images from the test dataset. In Table 4, the results of student classroom behavior detection for YOLOv8 and the improved version of YOLOv8 are demonstrated. The experimental results show that, for dense, mutual occlusion, and multi-scale targets, the improved YOLOv8 model outperforms the state-of-the-art model. It can reduce the leakage rate and improve the false detection rate, thus realizing effective detection, which basically meets the needs of the student classroom behavior detection task and has more practical application value.

## 5. Conclusions and Future Work

We conducted experiments on both the SCB-Dataset and the CrowdHuman dataset, and the results demonstrate a significant improvement in object detection accuracy, with mAP@0.5 increasing by 4.2% and 2.1%, respectively, compared to the original YOLOv8 model.

Our proposed student classroom behavior framework addresses challenges commonly found in classroom video imagery, such as density, mutual occlusion, and multi-scale scenarios. To tackle these challenges, we introduced a series of innovative methods. Firstly, we integrated the Res2Net module with the original C2f module in YOLOv8, creating a novel C2f_Res2block module. This module effectively handles multi-scale scenarios while enhancing model accuracy. Furthermore, we introduced the efficient multi-scale attention module (EMA) and the multi-head self-attention (MHSA) mechanism module, further bolstering the model’s performance. In the future, we will continue to focus on improving accuracy, reducing network parameters, and addressing the challenge of low-quality original video imagery through image enhancement techniques. These enhancements will contribute to further refining our student classroom behavior detection framework.

## Figures and Tables

**Figure 1 sensors-23-08385-f001:**
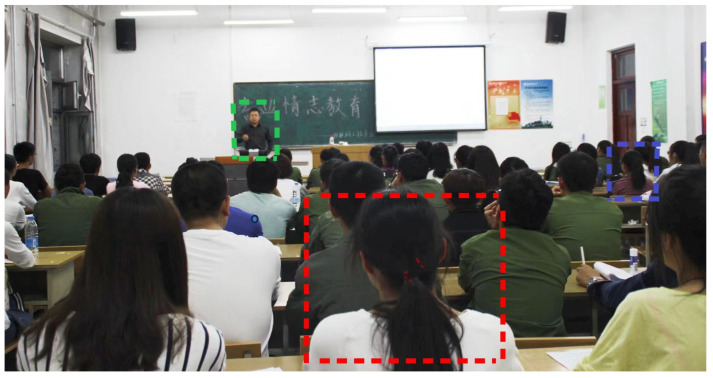
Classroom pictures.

**Figure 2 sensors-23-08385-f002:**
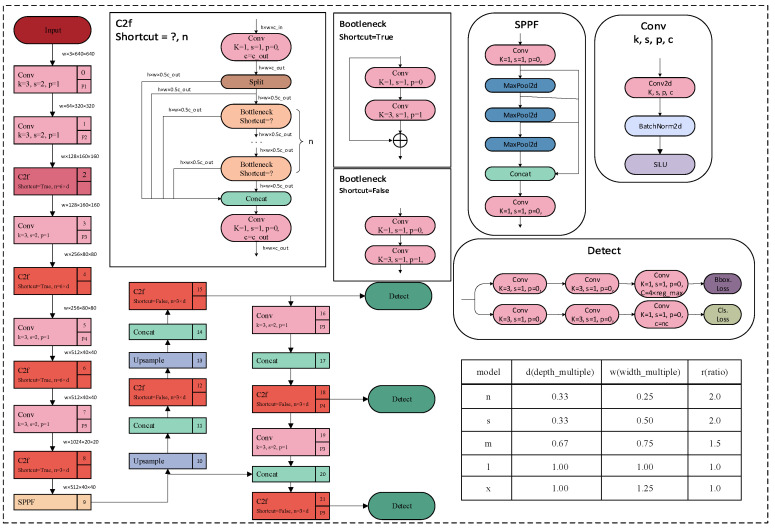
YOLOv8 structure diagram.

**Figure 3 sensors-23-08385-f003:**
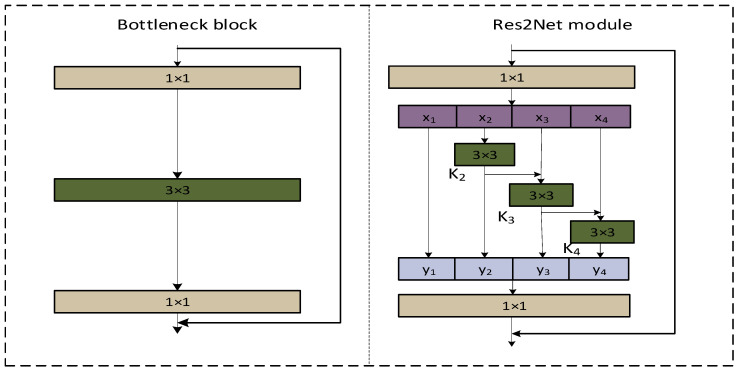
Comparison of bottleneck block and Res2Net module.

**Figure 4 sensors-23-08385-f004:**
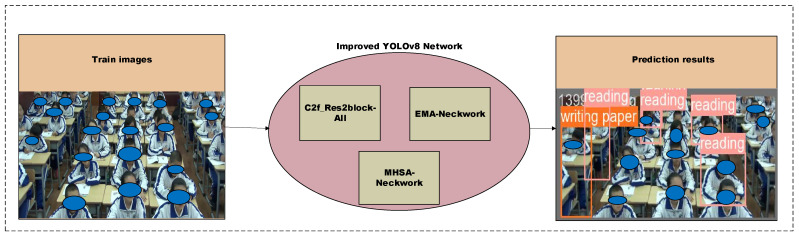
Overall framework for target detection.

**Figure 5 sensors-23-08385-f005:**
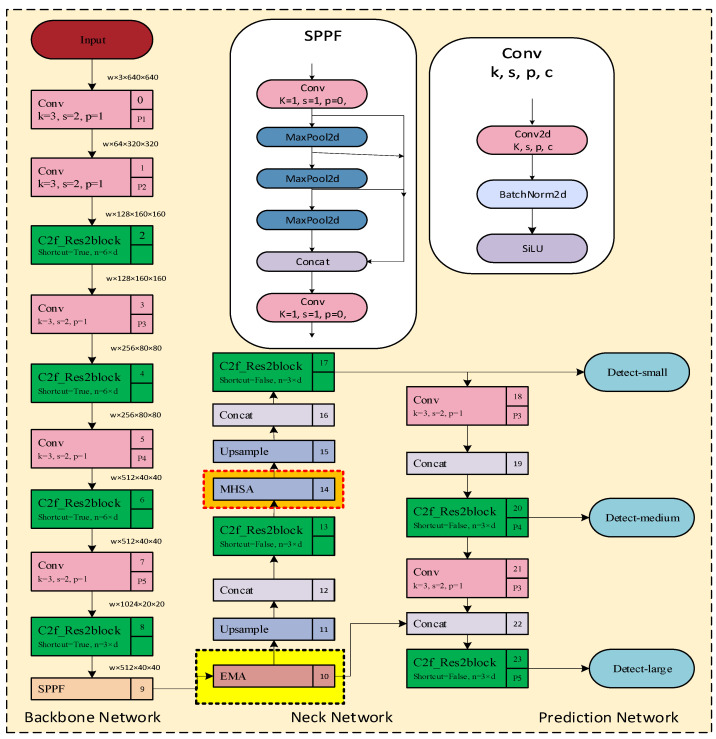
Improved YOLOv8 network framework.

**Figure 6 sensors-23-08385-f006:**
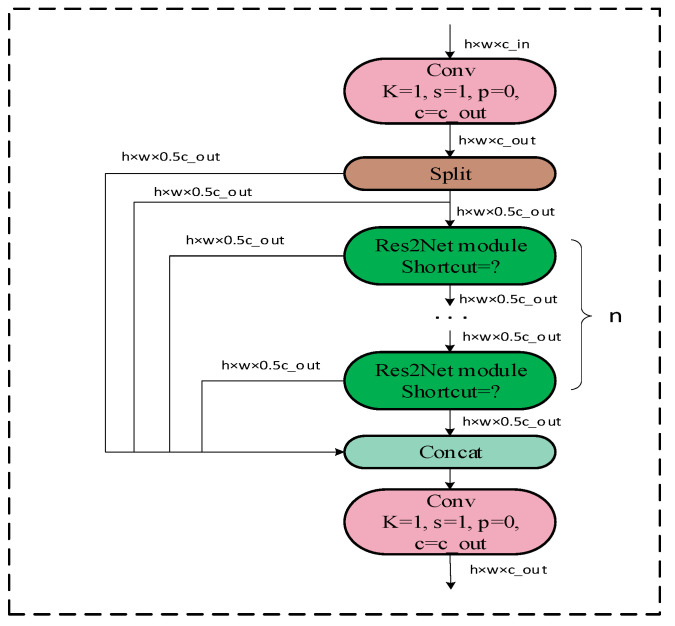
C2f_Res2block module.

**Figure 7 sensors-23-08385-f007:**
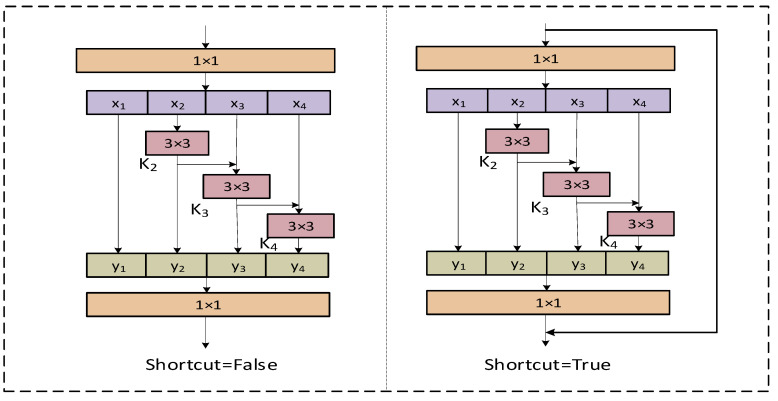
Res2Net module.

**Figure 8 sensors-23-08385-f008:**
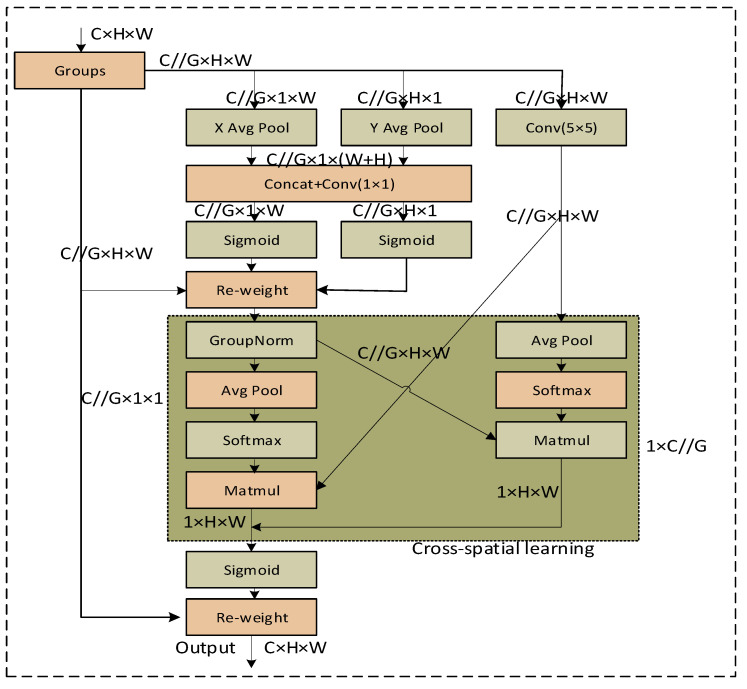
EMA module.

**Figure 9 sensors-23-08385-f009:**
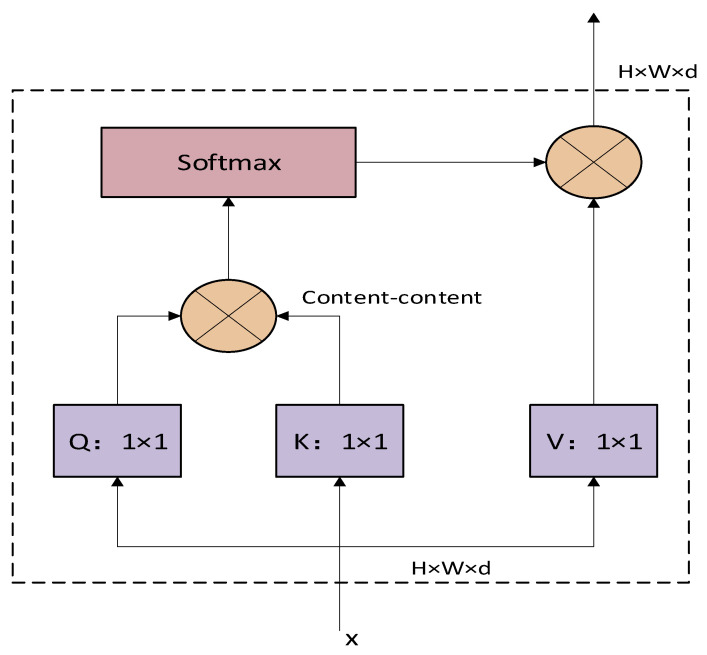
MHSA module.

**Figure 10 sensors-23-08385-f010:**
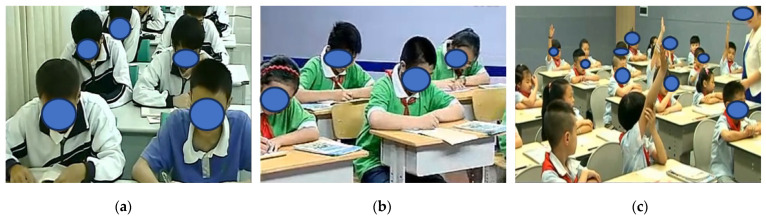
The SCB-Dataset comprises three categories, namely, (**a**) reading, (**b**) writing, and (**c**) raising hand.

**Figure 11 sensors-23-08385-f011:**
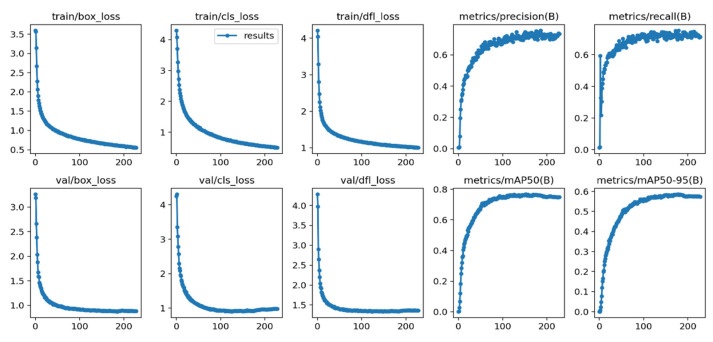
Performance values for the improved YOLOv8 model.

**Figure 12 sensors-23-08385-f012:**
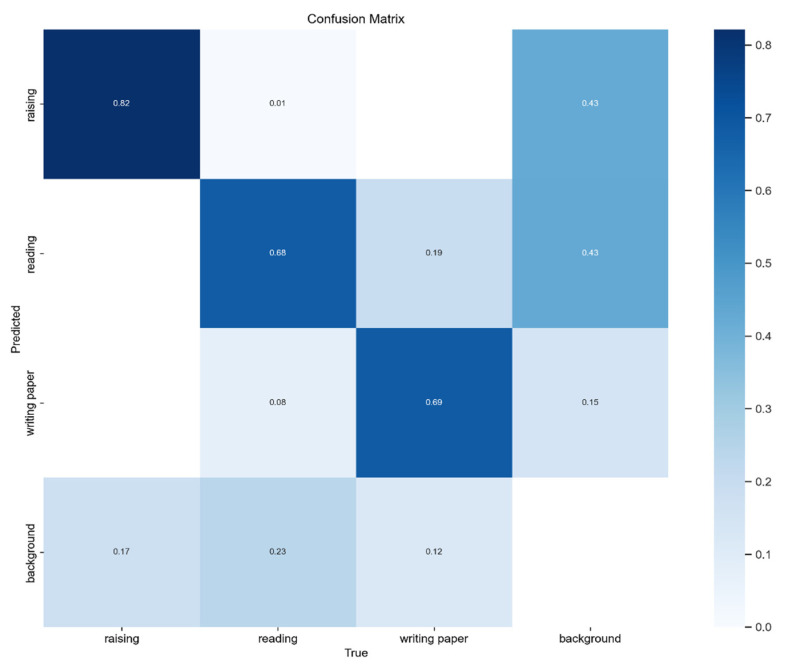
Confusion matrix for the proposed model.

**Figure 13 sensors-23-08385-f013:**
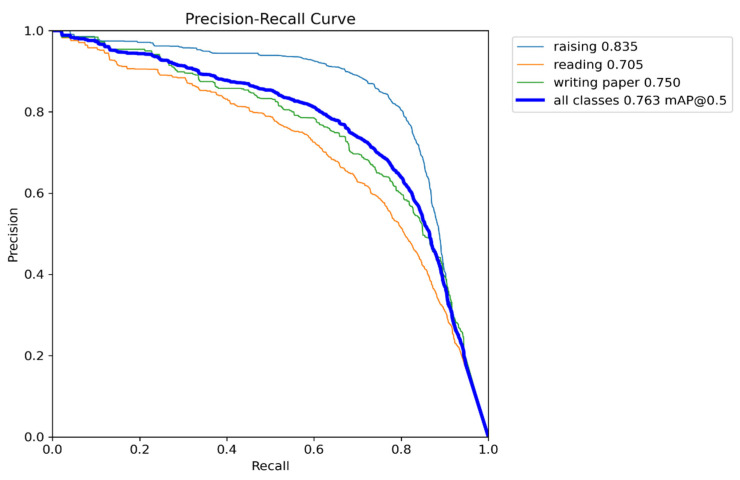
PR curves of the proposed model.

**Figure 14 sensors-23-08385-f014:**
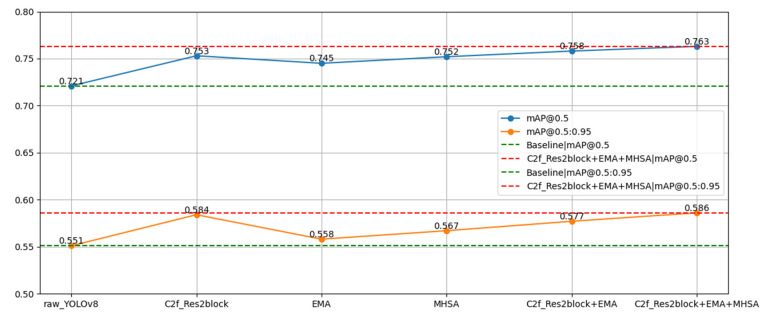
Improved-YOLOv8 ablation experiments result curve.

**Figure 15 sensors-23-08385-f015:**
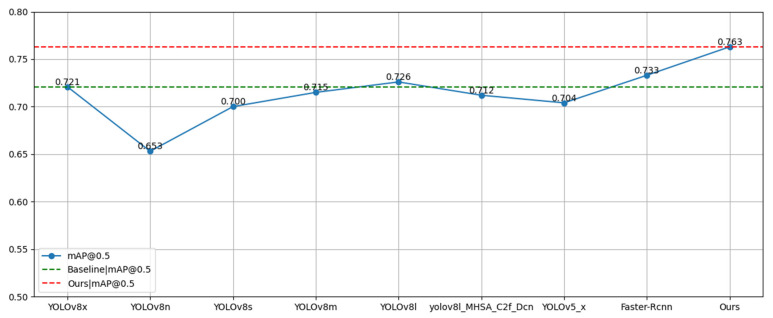
Comparison of experimental results—different models.

**Table 1 sensors-23-08385-t001:** Improved-YOLOv8 ablation experiments.

C2f_Res2block	EMA	MHSA	mAP@0.5	mAP@0.5:0.95
			0.721	0.551
√			0.753	0.584
	√		0.745	0.568
		√	0.752	0.567
√	√		0.758	0.577
√	√	√	0.763	0.586

**Table 2 sensors-23-08385-t002:** Comparison of training results on CrowdHuman dataset.

Model	Species	mAP@0.5	mAP@0.95
YOLOv8_row	Human	0.746	0.492
YOLOv8_improved	Human	0.767	0.512

**Table 3 sensors-23-08385-t003:** Comparative experiments—different models.

Method	AP			mAP@0.5
	Raising	Reading	Writing Paper	
YOLOv8x	0.822	0.645	0.696	0.721
YOLOv8n	0.766	0.59	0.603	0.653
YOLOv8s	0.801	0.63	0.669	0.7
YOLOv8m	0.815	0.643	0.687	0.715
YOLOv8l	0.825	0.649	0.704	0.726
YOLOv8l-MHSA-C2f-Cn	0.814	0.644	0.679	0.712
YOLOv5x	0.812	0.642	0.66	0.704
Faster-Rcnn	0.820	0.660	0.72	0.733
Ours	0.835	0.705	0.75	0.763

**Table 4 sensors-23-08385-t004:** Comparison of test results between YOLOv8 and the improved version of YOLOv8.

	Real Classroom	Before Improvement	After Improvement
False positive (8 cases of mistaking one student for multiple students)	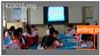	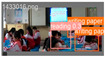	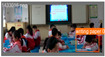
Missed detection (3 students’ learning statuses detected as 2 learning statuses)	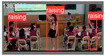	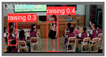	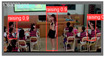
The targets are densely packed, and the obscuration effect is not good	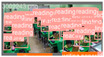	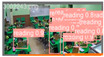	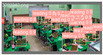
Low accuracy	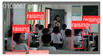	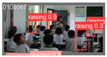	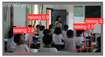
	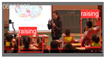	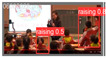	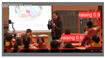

## Data Availability

The datasets generated during and/or analyzed during the current study are available from the corresponding authors upon reasonable request.

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
