# Peer review of "Student Behavior Detection in the Classroom Based on Improved YOLOv8"

_sensors, 2023, doi:10.3390/s23208385_

Round 1

Reviewer 2 Report

Overall, the papers are less innovative and are combinations of existing methods. The paper does not give an innovative algorithmic framework structure. In terms of experimental comparison, although the Yolov8 is compared, there is no comparison result with other networks that can do detection and classification tasks.

Reviewer 3 Report

Dear Authors

Thank you very much for allowing me to be part of the review of the paper entitled: Detection of Student Behavior in the Classroom Based on Enhanced YOLOv8. The paper is well structured according to the parameters of the journal.  The research topic is very successful, I would like to share with you some observations to improve the work done.  I will divide the comments into sections related to your work:

Introduction: The introduction includes current references related to the research and it is relevant to include the objective pursued, as it provides clarity to those interested in the topic.

It would be good to specify the main objective of the research at the end of the introduction.

It would be good to indicate whether Figure 1 is the author's own elaboration or the source.

 Materials and methods

The proposed methods are adequate

It would be good to review equation 1

Discussion: It would be good to improve the Discussion a little by including other similar works that help to test the hypotheses raised.

 Results: The results are presented adequately and clearly.  It would be good to include a discussion section of the results obtained and compare them with other similar works.

Conclusions:  It would be good  include in the conclusion a paragraph indicating how the objective has been achieved?

I do not feel qualified to give an opinion on the English language.

Reviewer 4 Report

1. How does the improved YOLOv8 model compare with other state-of-the-art detection models in terms of performance on the classroom dataset?

2. What are the computational costs of the improved model compared to the original YOLOv8? Is it significantly slower, or does it require more resources?

3. How does the improved model perform on datasets other than the SCB-Dataset? Is it generalizable to other classroom settings or even other object detection tasks?

4. What specific image augmentation techniques are being considered to address the challenge of low-quality video images? How might these augmentations enhance the model's robustness or performance?

5.  How was the SCB-Dataset split for training and validation? Were there any external datasets used for validation to ensure the model's generalizability?

6.What are the acknowledged limitations of the proposed model, and how might they be addressed in future iterations?

7. Is there a point of diminishing returns when adding more modules? Would adding further modifications continue to improve accuracy, or might it lead to overfitting or other issues?

8. If computational resources were limited and only one module could be incorporated into YOLOv8, which module provides the most significant standalone improvement?

Ok 

Round 2

Reviewer 2 Report

I think the authors still have not revised the manuscript accordingly to the comments. The network structure has not been optimized in any way, and the experiments have not been compared with other recent methods, just a new database has been added.

Reviewer 4 Report

1. How robust is the improved model to variations in lighting, camera angles, and classroom arrangements? Were any specific tests conducted to assess the model's robustness under different conditions?

Ok
